# Applying Foil Queue Microelectrode with Tapered Structure in Micro-EDM to Eliminate the Step Effect on the 3D Microstructure’s Surface

**DOI:** 10.3390/mi11030335

**Published:** 2020-03-24

**Authors:** Bin Xu, Kang Guo, Likuan Zhu, Xiaoyu Wu, Jianguo Lei

**Affiliations:** Guangdong Provincial Key Laboratory of Micro/Nano Optomechatronics Engineering, Shenzhen University, Shenzhen 518060, China; binxu@szu.edu.cn (B.X.); gk_szu@163.com (K.G.); zhulikuan@szu.edu.cn (L.Z.); wuxy@szu.edu.cn (X.W.)

**Keywords:** foil queue microelectrode, micro-EDM, step effect, tapered structure

## Abstract

When using foil queue microelectrodes (FQ-microelectrodes) for micro electrical discharge machining (micro-EDM), the processed results of each foil microelectrode (F-microelectrode) can be stacked to construct three-dimensional (3D) microstructures. However, the surface of the 3D microstructure obtained from this process will have a step effect, which has an adverse effect on the surface quality and shape accuracy of the 3D microstructures. To focus on this problem, this paper proposes to use FQ-microelectrodes with tapered structures for micro-EDM, thereby eliminating the step effect on the 3D microstructure’s surface. By using a low-speed wire EDM machine, a copper foil with thickness of 300 μm was processed to obtain a FQ-microelectrode in which each of the F-microelectrodes has a tapered structure along its thickness direction. These tapered structures could effectively improve the construction precision of the 3D microstructure and effectively eliminate the step effect. In this paper, the effects of the taper angle and the number of microelectrodes on the step effect were investigated. The experimental results show that the step effect on the 3D microstructure’s surface became less evident with the taper angle and the number of F-microelectrodes increased. Finally, under the processing voltage of 120 V, pulse width of 1 μs and pulse interval of 10 μs, a FQ-microelectrode (including 40 F-microelectrodes) with 10° taper angle was used for micro-EDM. The obtained 3D microstructure has good surface quality and the step effect was essentially eliminated.

## 1. Introduction

Micro electrical discharge machining (micro-EDM) is a non-contact machining technology, which has the advantage of a small cutting force. In view of this advantage, micro-EDM has been widely used in the processing of micro structures [1,2].

For achieving three-dimensional (3D) micro-EDM, Yu et al. [3] proposed the uniform wear method (UWM) and applied it to process 3D micro-cavities through layer-by-layer micro-EDM of a micro-electrode with a simple cross-section. In order to process complex 3D structures with arbitrary components, Rajurkar et al. [4] combined UWM and the computer-aided design (CAD)/computer-aided manufacturing (CAM) system in micro-EDM. With the purpose of investigating the effects of alternating-current on the energy usage and the erosion efficiency in the micro-EDM process, Yang et al. [5] built an electrical model and provided a theoretical analysis.

To improve the efficiency of 3D micro-EDM, Tong et al. [6] proposed the servo scanning 3D micro-EDM (3D SSMEDM) method based on the macro/micro-dual-feed spindle. In order to improve the machining efficiency and reduce the electrode wear, Fu et al. [7] proposed piezoelectric self-adaptive micro-EDM based on the inverse piezoelectric effect. To improve machining quality and machining efficiency of 3D micro-EDM, Yu et al. [8] proposed a new electrode wear compensation method, which combined the linear compensation method (LCM) with the uniform wear method (UWM). In order to fabricate micro groove arrays and columnar microstructures, Wu et al. [9] applied rotary dentate disc foil electrodes in micro-EDM in #304 stainless steel workpieces.

To improve machining accuracy of 3D micro-EDM, Nguyen et al. [10] identified and analyzed the error components of 3D micro-EDM milling process, which found that the corner radius of virtual electrode is also important to determine the machining accuracy. In order to achieve the high-precise machining of the micro rotating structure, Wang et al. [11] proposed micro reciprocated wire electrical discharge machining (wire-EDM) to fabricate micro-rotating structures. Using the low speed wire electrical discharge turning (LS-WEDT) method combined with the numerical control technology, Sun et al. [12] manufactured the microelectrodes and micro-cutting tools with good surface quality and high machining accuracy. To further study the discharge characteristics and machining mechanism of micro-EDM, Liu et al. [13] studied the variation trends of the discharge energy and discharge crater size in single-pulse experiments. Focus on the optimization of the processing parameters and quality control in micro-EDM, Bellotti et al. [14] applied a process fingerprint approach in micro-EDM drilling. In order to process micro-holes on tungsten carbide plates, D’Urso et al. [15] applied tubular electrodes in micro-EDM and evaluated the influence of variable process parameters on process performance. For achieving high precision machining of cubic boron nitride, Wyszynski et al. [16] described an application of EDM for drilling micro holes in cubic boron nitride and determined a set of parameters and technical specifications. Focus on the fabrication of deep micro-channels, Ahmed et al. [17] used wire-cut electrical discharge machining (EDM) to fabricate deep micro-channels with thin inter-channels fins.

Focus on the optimal selection of machining parameters, Świercz et al. [18] performed an analytical and experimental investigation of the influence of the EDM parameters. In order to understand the debris movement in high aspect ratio hole EDM machining, Liu et al. [19] developed a model to simulate the distribution and removal of debris in different machining conditions in ultrasonic assisted EDM with side flushing. To study the hydrogen dielectric strength forces in the EDM, António Almacinha et al. [20] applied electro-thermal model to simulate a single discharge in an electric discharge machining process. For obtaining an array micro-grooves, Wang et al. [21] developed a manufacturing method by applying disk electrode in micro electrochemical machining. For getting the high-efficiency removal, Zhang et al. [22] adopted a tool electrode with an optimized helical structure in tube electrode high-speed electrochemical discharge machining (TSECDM). Focus on the problem of the current micro-EDM pulse generator, Wang et al. [23] designed a micro-energy pulse source with narrow pulse width and high-voltage amplitude for getting more fine-etching ability. For fabricating micro punching mold with complex cross-sectional shape, Yu et al. [24] developed a micro punching system with a micro electrical discharge machining (EDM) module. To optimize the process parameters for micro EDM of Ti-6Al-4V alloy, Huang et al. [25] used the Taguchi method to determine the performance characteristics in micro EDM milling operations. In order to flush the debris generated in micro-EDM, Beigmoradi et al. [26] proposed a new numerical approach for enhancing flushing.

The above studies did good work on 3D micro-EDM and promoted the development of micro-EDM. For improving machining efficiency of 3D micro-EDM, Xu et al. [27,28] proposed a novel process to fabricate 3D micro-electrodes by superimposing multilayer 2D micro-structures and applied it in micro-EDM. However, the fabrication process of 3D micro-electrode is complicated and has a low success rate (30%).

Focusing on the complexity and low fabrication success rate of 3D microelectrodes, Xu et al. [29,30] discretized 3D micro-electrodes into several foil micro-electrodes and these foil micro-electrodes composed foil queue micro-electrode (FQ-microelectrode). According to the planned process path, each foil micro-electrode in FQ-microelectrode (Figure 1a) was sequentially applied in micro-EDM and processed results of each foil microelectrode (F-microelectrode) can be stacked to construct 3D microstructures. However, the surface of the 3D microstructure obtained from this process will have a step effect (Figure 1b), which will affect the surface quality and shape accuracy of the 3D microstructure. Focus on this problem, this paper used FQ-microelectrode with tapered structure for micro-EDM processing. These taper structures can effectively improve the construction accuracy of the 3D microstructure and effectively eliminate the step effect.

## 2. Method

Firstly, the 3D microstructure model was established using 3D modelling software. According to the 3D microstructure model, the corresponding 3D microelectrode model was obtained. Then, the 3D microelectrode model was sliced along its thickness direction to obtain a number of F-microelectrode models and thus machining data of each F-microelectrode can be obtained. Based on the machining data, the copper foil was processed by low-speed wire EDM machine to obtain each F-microelectrode and these F-microelectrodes composed the FQ-microelectrode.

FQ-microelectrode was applied for micro-EDM in sequence and processed results of each F-microelectrode can be stacked to construct 3D microstructures. Similar to the 3D printing process, the surface of 3D microstructure has a step effect (Figure 1), which seriously affects the surface quality and shape accuracy of the 3D microstructure.

When the electrode wear factor was not considered, the 3D microstructure obtained from micro-EDM of FQ-microelectrode was formed by many step superpositions. In this case, the step effect on the surface of the 3D microstructure was evident (Figure 2a). When the F-microelectrode had a tapered structure, the processing contour of the 3D microstructure was composed of oblique lines, which could better fit the design contour (Figure 2b), thereby reducing the step effect and improving the shape accuracy of the 3D microstructure.

## 3. Experimental Materials and Equipment

The FQ-microelectrode with taper structures was machined from a 300 μm thick copper foil using a LS-WEDM machine (Sodick company, Model: AP250LS, Suzhou, China), and then 3D micro-EDM was performed in cemented carbide. The FQ-microelectrode was observed by laser scanning confocal microscopy (Keyence company, model: VK-X250, Osaka, Japan). The surface topography of the 3D microstructure was observed by scanning electron microscopy (FEI company, Model: Quanta FEG 450, Hillsboro, OR, USA).

## 4. Experimental Results and Discussion

To eliminate the step effect on the 3D microstructure surface, this paper used FQ-microelectrode with tapered structures for micro-EDM to process 3D microstructure. This paper studied in detail the influence of different taper angles and numbers of F-microelectrodes on the step effect. Under the voltage of 72 V, the pulse width of 0.5 μs and the pulse interval of 5 μs, the FQ-microelectrodes were machined from copper foil with thickness of 300 μm by using the LS-WEDM machine.

### 4.1. Influence of Taper Angle on the Elimination of Step Effect

To study the effect of the taper angle on the step effect, FQ-microelectrodes with different taper angles were used for micro-EDM. Due to the limitations of the processing equipment, FQ-microelectrodes with a taper angle of more than 10° cannot be machined. Therefore, the taper angle of the FQ-microelectrodes was set to 0°, 2°, 4°, 6°, 8° and 10°. The workpiece material was cemented carbide, and the FQ-microelectrode containing 16 F-microelectrodes was fabricated in 300 μm thick copper foil. The processing object was 1/4 sphere with diameter of 600 μm. Based on the previous studies [17,18], under the processing voltage of 120 V, the pulse width of 1 μs and the pulse interval of 10 μs, the processing object fabricated by micro-EDM had well surface morphology.

The experimental results are shown in Figure 3. When the taper angle of the F-microelectrode is 0°, the number of steps on the 3D microstructure surface is highest and the step effect is evident (Figure 3a). When the taper angle of the F-microelectrode is 10°, the number of steps on the 3D microstructure surface is 2 and the step effect is not evident. When the F-microelectrode has a taper structure, the processing contour of the 3D microstructure is composed of oblique lines, which effectively improves the shape precision of the 3D microstructure and reduces the step effect. Therefore, as the taper angle of the F-microelectrode increases, the number of steps on the 3D microstructure surface gradually decreases and the step effect becomes increasingly less evident.

To further clarify the position of the steps, the cross-sectional profile of the 3D microstructure was measured by laser confocal microscopy and the experimental results are shown in Figure 4.

When the taper angle of the F-microelectrode is 0°, the steps are mainly distributed in the middle and tail of the spherical surface. When the taper angle of the F-microelectrode gradually increases from 0° to 10°, the steps in the middle of the spherical surface are gradually eliminated, and the steps in the tail of the spherical surface are somewhat attenuated. When the F-microelectrode had a tapered structure, the processing contour of the 3D microstructure was composed of oblique lines. If the slope of the oblique line was close to the slope of the processing results of the adjacent F-microelectrodes, the steps on this position can be substantially eliminated. From the processing position of the first F-microelectrode to the last F-microelectrode, the height difference of the processing results of the adjacent F-microelectrodes is continuously increased (Figure 5), so the slope of the processing results of the adjacent F-microelectrodes is continuously increased. Therefore, when the taper angle of the F-microelectrode is close to the slope of the processing results of the adjacent F-microelectrodes, the step effect can be effectively eliminated (Figure 5). Thus, in the middle of the spherical surface, when the taper angle of the F-microelectrode is 10°, the taper angle is relatively close to the slope of the processing results of the adjacent F-microelectrodes. Therefore, the steps on this position is substantially eliminated. In the tail of the sphere surface, the taper of the F-microelectrode differs greatly from the slope of the processing results of the adjacent F-microelectrodes, which results in a more pronounced step at that location. 

### 4.2. Influence of Numbers of F-Microelectrode on the Elimination of Step Effect

Due to the limitation of processing equipment, FQ-microelectrodes with a taper angle of more than 10° cannot be machined. Therefore, it is difficult to eliminate the steps in the tail of the sphere surface. To focus on this problem, this paper proposes to eliminate the steps in the tail of the sphere surface by reducing slice thickness of the 3D microelectrode model and thereby increasing the number of the F-microelectrodes.

To study the effect of the number of F-microelectrodes on the step effect, FQ-microelectrodes with different numbers of F-microelectrodes were applied for micro-EDM. The FQ-microelectrodes had 16, 25, 33 and 40 F-microelectrodes. The workpiece material was cemented carbide. The FQ-microelectrode was fabricated in 300 μm thick copper foil and every F-microelectrode had a taper angle of 10°. The processing object was 1/4 sphere with diameter of 600 μm, the processing voltage was 120 V, the pulse width was 1 μs and the pulse interval was 10 μs.

As shown in Figure 6, when the number of F-microelectrodes is 16, the 3D microstructure surface has a small number of steps and these steps are located in the tail of the sphere surface. As the number of F-microelectrodes increasing, the steps on the 3D microstructure surface become fewer and fewer. When the number of F-microelectrodes increases to 40, the steps on the 3D microstructure surface are essentially eliminated (Figure 6d). These experimental results prove that the step effect of the 3D microstructure surface can be effectively eliminated by increasing the number of the F-microelectrodes. To further clarify the position of the steps, the cross-sectional profile of the 3D microstructure was measured by laser confocal microscopy and the experimental results are shown in Figure 7.

As shown in Figure 7, when the number of F-microelectrodes is 16, the step is mainly distributed at the tail of the spherical surface. With the number of F-microelectrodes gradually increasing from 16 to 40, the position at which the steps occur gradually moves to the tail of the sphere, until it is eliminated. As slice thickness of the 3D microelectrode model decreasing, the number of F-microelectrodes will increase, which could improve the fitting precision of 3D microstructure. Therefore, in the tail of the sphere, the step can be eliminated through increasing the number of F-microelectrodes. In addition, during the micro-EDM, the wear of the F-microelectrode is unavoidable. Under the effect of micro-EDM, the vertical angle at the end face of the F-microelectrode is worn and becomes rounded corner. Under the influence of these factors, the surface of the processing results is processed into a corresponding curved surface and thus the steps on the 3D microstructure surface are gradually eliminated.

## 5. Application of FQ-Microelectrode with Tapered Structures in Micro-EDM

To verify the feasibility of the proposed process, an FQ-microelectrode was prepared using a 300 μm thick copper foil. The FQ-microelectrode contained 40 F-microelectrodes, each of which had taper structure with taper angle of 10° (Figure 8a). The process parameters of the FQ-microelectrode preparation were described as follows: wire cutting voltage of 72 V, the pulse width of 0.5 μs and the pulse interval of 5 μs. The FQ-microelectrode was applied in micro-EDM and its process parameters were described as follows: pulse width of 1 μs, pulse interval of 10 μs and voltage of 120 V. The processing object was a hemisphere with diameter of 600 μm (Figure 9) and the workpiece material was cemented carbide.

The 3D microstructure was observed and measured by using scanning electron microscopy and laser scanning confocal microscopy. The experimental results are shown in Figure 8 and Table 1. From the experimental results, it can be seen that the surface quality of the 3D microstructure is good (Figure 8b) and the surface step effect is eliminated (Figure 8c). The dimensional accuracy of 3D microstructure is well and the maximum dimensional error is within 10 μm.

## 6. Conclusions

Using FQ-microelectrodes for micro-EDM, processed results of each F-microelectrode can be stacked to construct 3D microstructures. However, the surface of the 3D microstructure obtained from this process will have step effect, which will affect the surface quality and shape accuracy of the 3D microstructure. To focus on this problem, this paper proposed to eliminate the step effect by using FQ-microelectrodes with tapered structures for micro-EDM and increasing the number of F-microelectrodes. Through the detailed study, the following conclusions can be drawn:(1)When the FQ-microelectrodes with tapered structures were used for micro-EDM, the step effect on the surface of the 3D microstructure can be significantly weakened and the shape accuracy can be improved.(2)By reducing the slice thickness of the 3D microelectrode model and thereby increasing the number of F-microelectrodes, the step effect on the 3D microstructure surface can be further eliminated. When the taper angle of the F-microelectrode is 10° and the number of F-microelectrodes is 40, the step effect on the surface of the 3D microstructure can be essentially eliminated.(3)Under the processing voltage of 120 V, pulse width of 1 μs and pulse interval of 10 μs, the FQ-microelectrode (including 40 F-microelectrodes) with 10° taper angle was used for micro-EDM. The obtained 3D microstructure has good surface quality and the step effect was essentially eliminated. The dimensional accuracy of 3D microstructure is well and the maximum dimensional error is within 10 μm.

## Figures and Tables

**Figure 1 micromachines-11-00335-f001:**
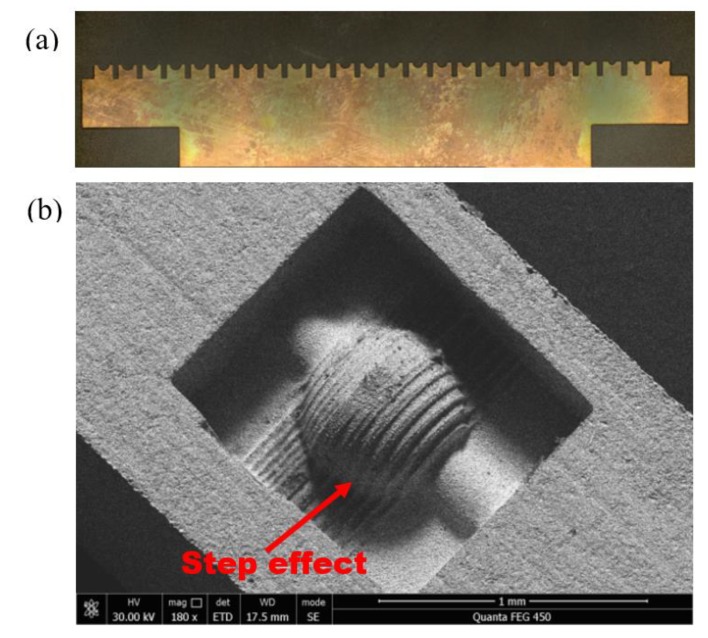
(**a**) Foil queue (FQ)-microelectrode; (**b**) 3D microstructure fabricated by the micro-electrical discharge machining (EDM) of FQ-microelectrode.

**Figure 2 micromachines-11-00335-f002:**
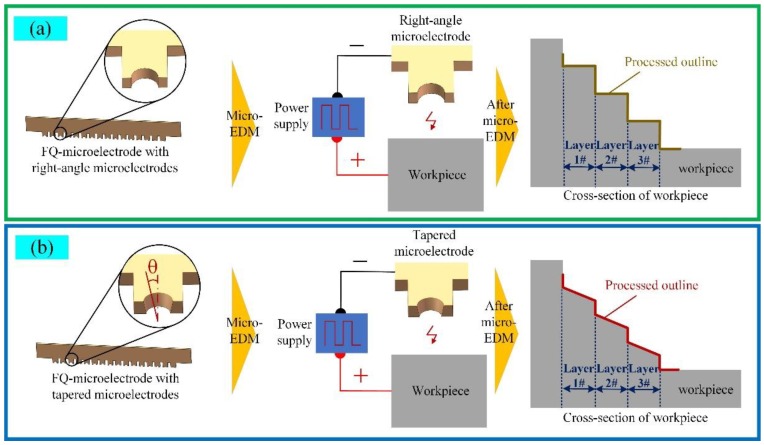
(**a**) Fabricating 3D microstructure based on the FQ-microelectrode without tapered structure; (**b**) Fabricating 3D microstructure based on the FQ-microelectrode with tapered structure.

**Figure 3 micromachines-11-00335-f003:**
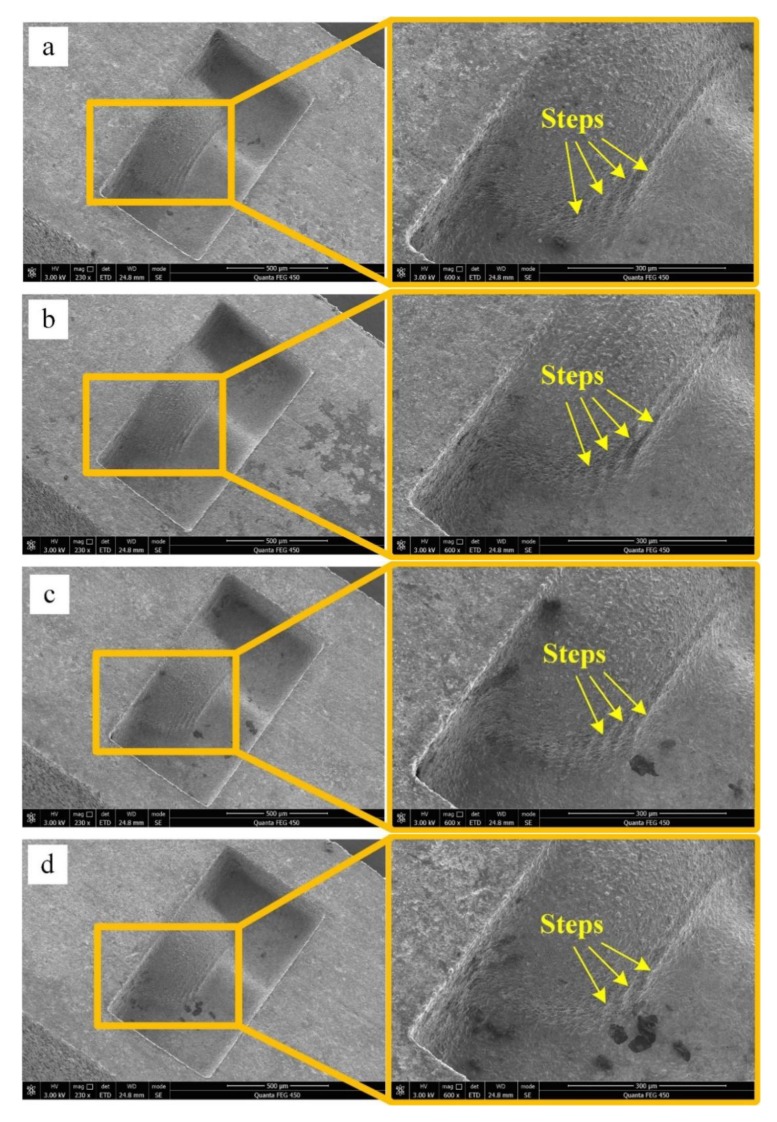
Micro-EDM results of FQ-microelectrodes with different taper angles: (**a**) 0°; (**b**) 2°; (**c**) 4°; (**d**) 6°; (**e**) 8°; (**f**) 10°.

**Figure 4 micromachines-11-00335-f004:**
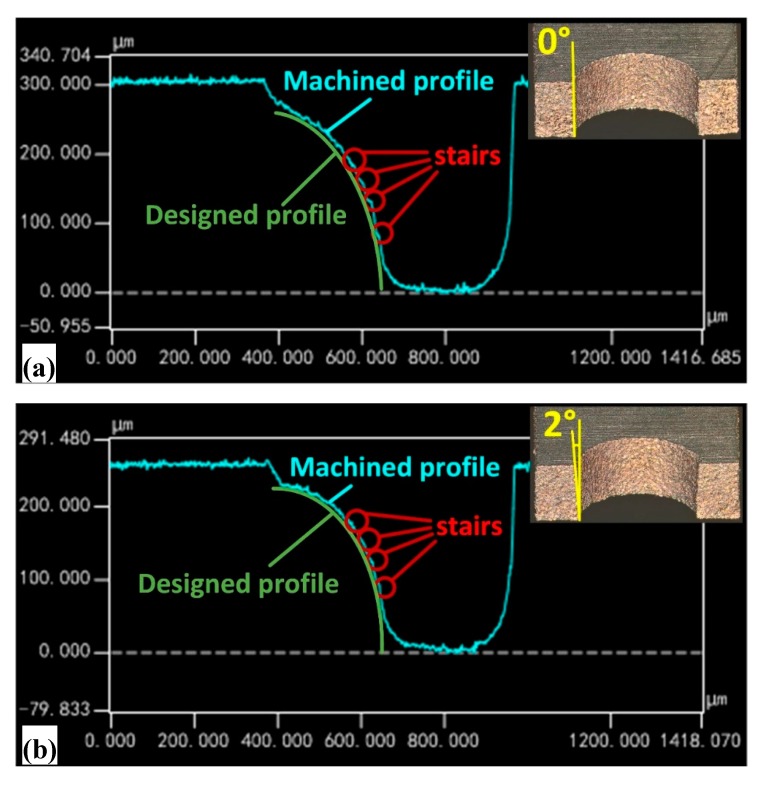
Micro-EDM results of FQ-microelectrode with different tapers observed by laser scanning confocal microscopy: (**a**) 0°; (**b**) 2°; (**c**) 4°; (**d**) 6°; (**e**) 8°; (**f**) 10°.

**Figure 5 micromachines-11-00335-f005:**
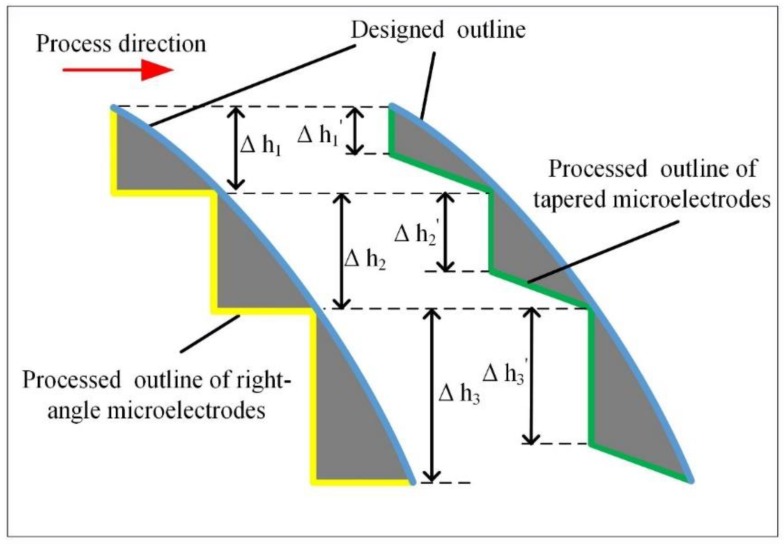
The height difference of the processing results of the adjacent F-microelectrodes.

**Figure 6 micromachines-11-00335-f006:**
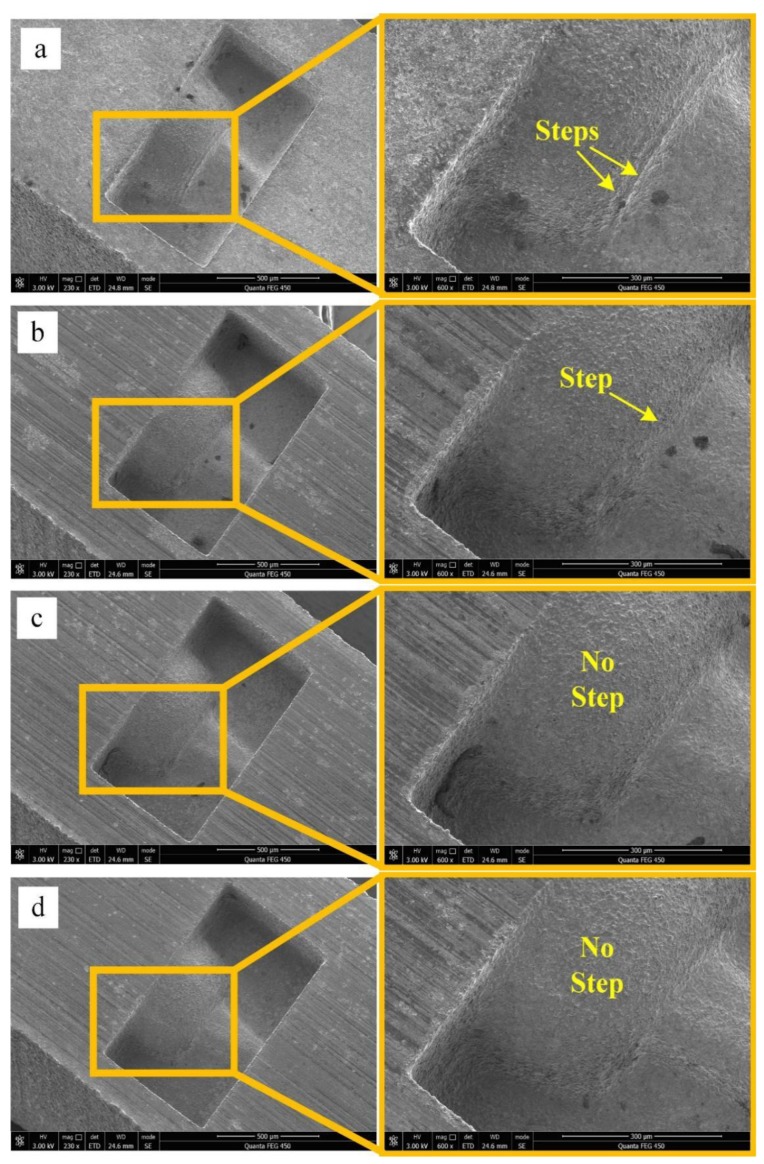
Micro-EDM results of FQ-microelectrodes with different number of F-microelectrodes: (**a**) 16; (**b**) 25; (**c**) 33; (**d**) 40.

**Figure 7 micromachines-11-00335-f007:**
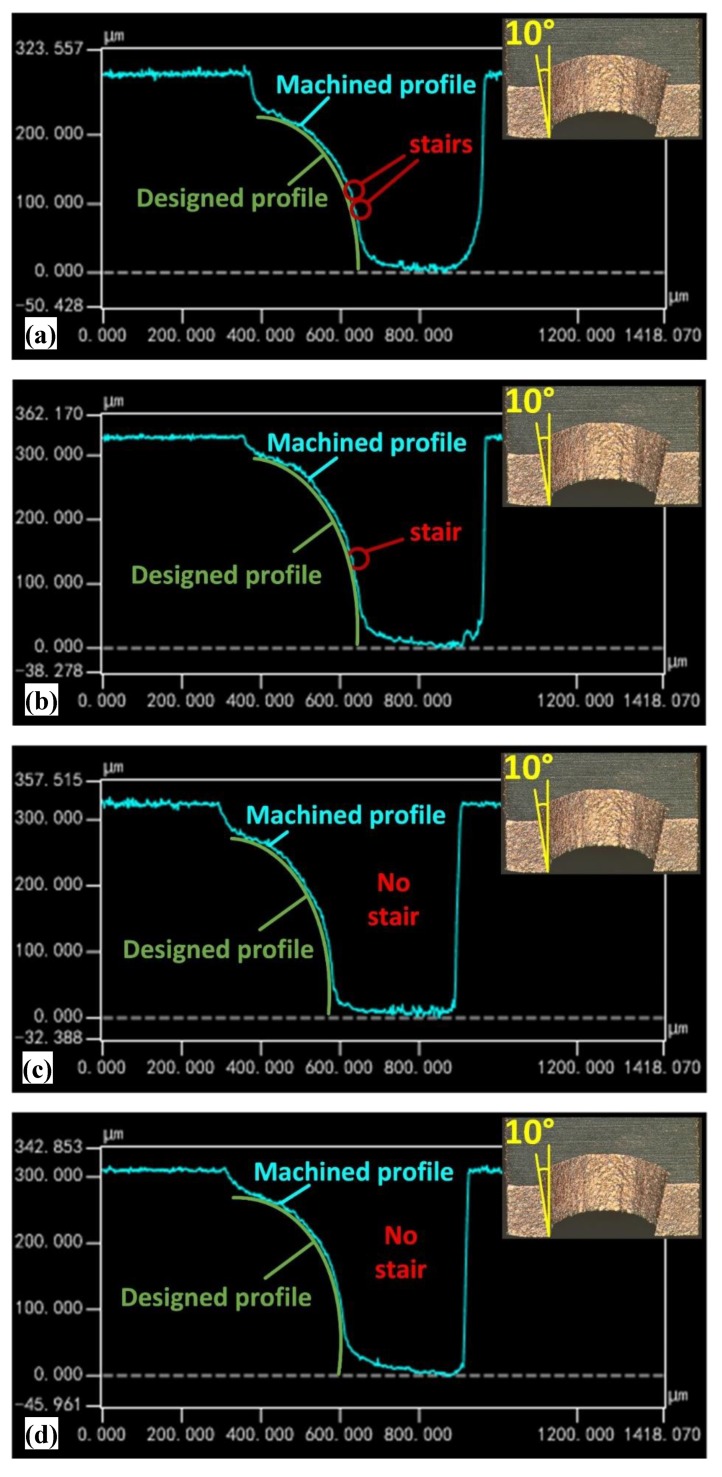
Micro-EDM results of FQ-microelectrodes with different number of F-microelectrodes observed by laser scanning confocal microscopy: (**a**) 16; (**b**) 25; (**c**) 33; (**d**) 40.

**Figure 8 micromachines-11-00335-f008:**
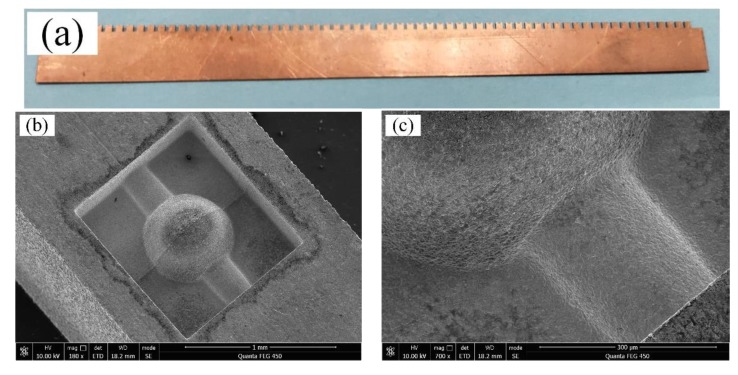
(**a**) FQ-microelectrode with tapered structures; (**b**,**c**) Micro-EDM results of FQ-microelectrodes observed by scanning electron microscopy.

**Figure 9 micromachines-11-00335-f009:**
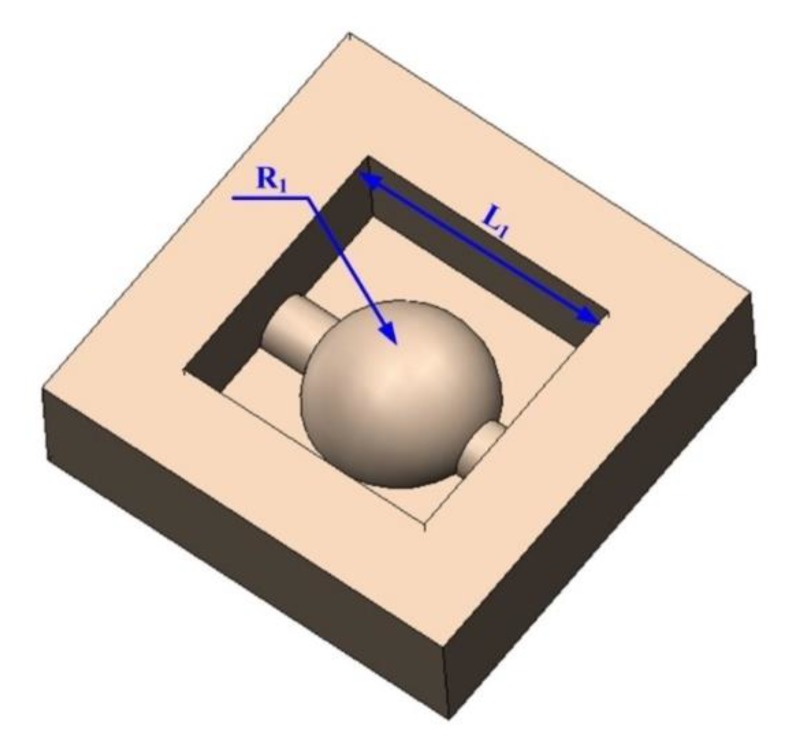
The computer-aided design (CAD) model of 3D microstructure.

**Table 1 micromachines-11-00335-t001:** Dimensional comparison between computer-aided design (CAD) model shown in Figure 9 and micro electrical discharge machining (micro-EDM) result shown in Figure 8.

Dimensional Symbols	Dimensional Values (μm)			Processing Time
	CAD Model Shown in Figure 9	Micro-EDM Result Shown in Figure 8	Errors	
L1	1156	1150	6	120 min
R1	300	295	5	
Depth	330	322	8	

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
