# Peer review of "Applying Foil Queue Microelectrode with Tapered Structure in Micro-EDM to Eliminate the Step Effect on the 3D Microstructure’s Surface"

_micromachines, 2020, doi:10.3390/mi11030335_

Round 1

Reviewer 1 Report

Review of micromachines-746629-peer-review-v1: “Applying foil queue microelectrode with tapered structure in micro-EDM to eliminate the step effect on the 3D microstructure’s surface”

 The subject of the paper is extremely relevant with the topics of the journal.

Its significance with respect to their industrial value is great and deals with a research area that attracts a lot of attention.

The references are well selected and well-presented within the paper, while at the same time cover the latest development in the area. The paper is well structured and the aims are very clear and gradually presented. A number of different tapers and different number of F-microelectrodes are used interpedently. At the end, the verification of the outcome is applied on a microstructure and proves the validity of the research.

Although the paper is of a high quality, I would suggest the authors to incorporate two minor changes:

  • In fig. 3, the values of the taper angles used should be incorporated within the graphics for each case
  • In fig. 6 and 7, the values of the number of F-microelectrodes should be incorporated within the graphics for each case.

In such cases, the presentation of the figures becomes independent of the figure description.

My proposal to the editor is to accept the paper with minor revisions

Author Response

1. In fig. 3, the values of the taper angles used should be incorporated within the graphics for each case. Author response: Thanks for your suggestion and apologize for our negligence. Based on your comment, we have added the values of the taper angles in fig. 3. Thanks for your useful comment again. 2. In fig. 6 and 7, the values of the number of F-microelectrodes should be incorporated within the graphics for each case. Author response: Thanks for your suggestion and apologize for our negligence. Based on your comment, we have added the values of the number of F-microelectrodes in fig. 6 and fig. 7. Thanks for your useful comment again.

Reviewer 2 Report

The paper is focused on the opportunity to use tapered foil microelectrodes as solution to remove the step effect from 3D mnicrostructure micro-EDM machininig.

Although the paper sounds interesting at first glance, the English language, the abstract, the reported state of the art, the organization of the paper, the reported figures and results require heavy revisions.

In the present form I do not recommend this paper as suitable for publication.

Please, take note of the following comments and suggestions for further revision and resubmission, prior modifications to the overall manuscript.

English: no particular grammar mistakes are present. Nonetheless, the English language is somehow very inappropriate for a scientific paper. Lots of repetitive sentences are found throughout the manuscript. Please, improve the way the authors report every concept, in particular concerning the explanations about their results. Otherwise, it is not possible to catch the real novelty introduced by the present work.

Abstract: it sounds repetitive. It does not provide a deep insight about the novelty of the paper.

Introduction-state of the art: This section is not appropriate to the main topic claimed by the authors. The first part related to ref. 1-15 is just a list of some papers concerning micro-EDM in the widest point of view. Only 3 references match the claimed issue, namely “solving step effect on micro-EDMed  structure surfaces”. Therefore, if the authors decide to keep the reported 18 references, it is mandatory that they must include in the introduction the results of the state of the art in relation to their proposed solution: this is essential in order to have a clear view of their novel contribution with respect to previous papers. Moreover, the suitable references are by the authors selves (ref 17-18). However, also in this case, the authors made no effort to underline what is the novelty introduced by the present paper compared to their previous ones.

Section 2: Method

This section should be intended to report the machining of FQ foil microelectrode. However, the way the FQ microelectrodes are obtained by wire EDM is not explained. Several results about their fabrication are reported later on in the manuscript. Figure 2 doesn’t help to clarify the procedure followed by the authors in micro-EDM machine of the microelectrodes and the subsequent micro-EDM of the final structure.

Section 3: the micro-EDM approach used for the fabrication of the final structure is not specified at all. It is important to have an indication of the approach among milling, drilling or sinking. Additionally, all information about micro-EDM process parameters, both regarding wire EDM and the approach used to machine the final structure, must be reported in this part. The only information concern V, W and T, but it is not clear how the authors decided to use such process parameters.

Section 4: the first two sentences opening the subsections 4.1 4.2 is completely unnecessary. Section 4.1: what do the authors mean by “due to limitations”? What kind of limitations? Figures 3 have very poor resolution: therefore it is quite complicated to follow and agree with the discussion about the presence or reduction of step effects in the final structures. Moreover, several times the authors use the word “obvious”: I warmly suggest to replace this word with “evident”. Line 139: “increasing” I suppose it is “increases”. From line 141: the results concerning the profile of FQ electrodes must be reported before the analysis of the surface of the final structure. Line 157-159: the sentence “from the processing…” is by no means clear. Please rephrase it so that the reader can catch the real meaning. What do the authors mean by saying “processing results”? line 161-163: please rephrase the sentence “thus in the middle..”. It’s quite obscure to the reader. Line 214-216: “Under the influence…”  In relation to this sentence: how could the authors be sure that the elimination of the steps was not ascribed to tool wear? I suggest the authors to verify this issue.

Section 5: Figure5a is too small. Table 1: the dimensions of the microstructures must be defined in the text as well: L1, R1 and depth are not defined elsewhere.

Edit: please, choose between “F-electrodes or FQ electrodes”. Uniformity is required to bring clarity to the manuscript.

Author Response

1. English: no particular grammar mistakes are present. Nonetheless, the English language is somehow very inappropriate for a scientific paper. Lots of repetitive sentences are found throughout the manuscript. Please, improve the way the authors report every concept, in particular concerning the explanations about their results. Otherwise, it is not possible to catch the real novelty introduced by the present work.

Author response: Thanks for your suggestion. Based on your comment, improvements of English in the paper were made and revised portion are marked in red in the paper. Thanks for your useful comment again.

2. Abstract: it sounds repetitive. It does not provide a deep insight about the novelty of the paper.

Author response: Thanks for your suggestion. Using foil queue microelectrode (FQ-microelectrode) for micro electrical discharge machining (micro-EDM), processed results of each foil microelectrode (F-microelectrode) can be stacked to construct three-dimensional (3D) microstructure. However, the surface of the 3D microstructure obtained from this process will have a step effect, which has an adverse effect on the surface quality and shape accuracy of the 3D microstructure. Focus on this problem, this paper proposed to use FQ-microelectrode with tapered structure for micro-EDM, thereby eliminating the step effect on the 3D microstructure’s surface. By using low-speed wire EDM machine, a copper foil with thickness of 300 μm was processed to obtain a FQ-microelectrode in which each of the F-microelectrode has a tapered structure along its thickness direction. These tapered structures could effectively improve the construction precision of 3D microstructure and effectively eliminate the step effect. Based on your comment, we have made revision in the abstract and revised portion are marked in red in the paper. Thanks for your useful comment again.

3. Introduction-state of the art: This section is not appropriate to the main topic claimed by the authors. The first part related to ref. 1-15 is just a list of some papers concerning micro-EDM in the widest point of view. Only 3 references match the claimed issue, namely “solving step effect on micro-EDMed structure surfaces”. Therefore, if the authors decide to keep the reported 18 references, it is mandatory that they must include in the introduction the results of the state of the art in relation to their proposed solution: this is essential in order to have a clear view of their novel contribution with respect to previous papers. Moreover, the suitable references are by the authors selves (ref 17-18). However, also in this case, the authors made no effort to underline what is the novelty introduced by the present paper compared to their previous ones.

Author response: Thanks for your suggestion. The references in this paper are all related to micro-EDM and it is impossible for every reference to be closely related to the topic of this paper. The purpose and method of each reference in the field of micro-EDM were discussed. In reference [17] and [18], Xu et al. discretized 3D micro-electrodes into several foil micro-electrodes and these foil micro-electrodes composed foil queue micro-electrode (FQ-microelectrode). According to the planned process path, each foil micro-electrode in FQ-microelectrode (as shown in Fig. 1a of the manuscript) was sequentially applied in micro-EDM and processed results of each foil microelectrode (F-microelectrode) can be stacked to construct 3D microstructure. However, the surface of the 3D microstructure obtained from this process will have a step effect (as shown in Fig. 1b of the manuscript), which will affect the surface quality and shape accuracy of the 3D microstructure. Focus on this problem, this paper used FQ-microelectrode with tapered structure for micro-EDM processing. These taper structures can effectively improve the construction accuracy of the 3D microstructure and effectively eliminate the step effect.

Based on your comment, we have made revision in the introduction to have a clear view of their novel contribution and revised portion are marked in red in the paper. Thanks for your useful comment again.

 4. Section 2: Method

This section should be intended to report the machining of FQ foil microelectrode. However, the way the FQ microelectrodes are obtained by wire EDM is not explained. Several results about their fabrication are reported later on in the manuscript. Figure 2 doesn’t help to clarify the procedure followed by the authors in micro-EDM machine of the microelectrodes and the subsequent micro-EDM of the final structure.

Author response: Thanks for your suggestion. Firstly, the 3D microstructure model was established using 3D modelling software. According to the 3D microstructure model, the corresponding 3D microelectrode model was obtained. Then, the 3D microelectrode model was sliced along its thickness direction to obtain a number of F-microelectrode models and thus machining data of each F-microelectrode can be obtained. Based on the machining data, the copper foil was processed by low-speed wire EDM machine to obtain each F-microelectrode and these F-microelectrodes composed the FQ-microelectrode. Figure 2 illustrated that the F-microelectrode with tapered structure can effectively reduce the step effect on the surface of 3D microstructure.

Based on your comment, we have made revision in section 2 to describe the machining process of the FQ-microelectrode and revised portion are marked in red in the paper. Thanks for your useful comment again.

5. Section 3: the micro-EDM approach used for the fabrication of the final structure is not specified at all. It is important to have an indication of the approach among milling, drilling or sinking. Additionally, all information about micro-EDM process parameters, both regarding wire EDM and the approach used to machine the final structure, must be reported in this part. The only information concern V, W and T, but it is not clear how the authors decided to use such process parameters.

Author response: Thanks for your suggestion. FQ-microelectrode was applied for micro-EDM in sequence and processed results of each F-microelectrode can be stacked to construct 3D microstructures. Based on your comment, we have made revision in section 2 to describe the fabrication of the final structure and revised portion are marked in red in the paper.

This paper focuses on the influence of taper structure of the F-microelectrode on the elimination of step effect. Process parameters for fabricating the FQ-microelectrode and applying it in micro-EDM are not the focus of this paper. Based on your comment, we have made revision in section 3 to describe the process parameters of micro-EDM and wire EDM. The revised portion are marked in red in the paper. Thanks for your useful comment again.

6. Section 4: the first two sentences opening the subsections 4.1 4.2 is completely unnecessary. Section 4.1: what do the authors mean by “due to limitations”? What kind of limitations? Figures 3 have very poor resolution: therefore it is quite complicated to follow and agree with the discussion about the presence or reduction of step effects in the final structures. Moreover, several times the authors use the word “obvious”: I warmly suggest to replace this word with “evident”. Line 139: “increasing” I suppose it is “increases”. From line 141: the results concerning the profile of FQ electrodes must be reported before the analysis of the surface of the final structure. Line 157-159: the sentence “from the processing…” is by no means clear. Please rephrase it so that the reader can catch the real meaning. What do the authors mean by saying “processing results”? line 161-163: please rephrase the sentence “thus in the middle..”. It’s quite obscure to the reader. Line 214-216: “Under the influence…” In relation to this sentence: how could the authors be sure that the elimination of the steps was not ascribed to tool wear? I suggest the authors to verify this issue.

Author response: Thanks for your suggestion. The first two sentences in section 4 illustrated the main work of section 4 and this was a general statement. “due to limitations” means that FQ-microelectrode with a taper angle of more than 10° can not be machined.

In this paper, the influence of taper angles on the elimination of step effect was studied. 3. When the taper angle of the F-microelectrode is 0°, the number of steps on the 3D microstructure surface is highest and the step effect is evident. When the taper angle of the F-microelectrode is 10°, the number of steps on the 3D microstructure surface is 2 and the step effect is not evident. Figures 3 can clearly represent the steps on the surface of 3D microstructure. Based on your comment, we have made revision in section 4.1 and revised portion are marked in red in the paper.

When the F-microelectrode has a taper structure, the processing contour of the 3D microstructure is composed of oblique lines, which effectively improves the shape precision of the 3D microstructure and reduces the step effect. This can explain the profile of FQ-microelectrodes.

The sentence “from the processing…” refers to the processing position of each F-microelectrode. Based on your comment, we have made revision in Line 157-159 and revised portion are marked in red in the paper.

“processing results” means that processed results of each F-microelectrode. Based on your comment, we have made revision in section 2 to explain the meaning of “processing results” and revised portion are marked in red in the paper.

When the F-microelectrode had a tapered structure, the processing contour of the 3D microstructure was composed of oblique lines. If the slope of the oblique line was close to the slope of the processing results of the adjacent F-microelectrodes, the steps on this position can be substantially eliminated. Based on your comment, we have made revision in Line 161-163 and revised portion are marked in red in the paper.

Figure 1 shows that the FQ microelectrode without tapered structure and its micro-EDM results. From figure 1, we can know that tool wear can reduce the step effect (Figure 1e and Figure 1f). However, tool wear can not eliminate the step effect completely. Therefore, applying FQ-microelectrode with tapered structure for micro-EDM and increasing the number of F-microelectrodes can eliminate the step effect completely. Thanks for your useful comment again.

7. Section 5: Figure5a is too small. Table 1: the dimensions of the microstructures must be defined in the text as well: L1, R1 and depth are not defined elsewhere.

Author response: L1, R1 and depth are defined in the Figure 8. Figure5a is too small and we apologize for this. Unfortunately, under the circumstances (2019 Novel Coronavirus), we are not advised to go back to college and suggested to work at home (including giving online lectures to students, scientific research and so on). Can we provide better Figure 5 in the future? Thanks for your useful comment and apologize for this again.

8. Edit: please, choose between “F-electrodes or FQ electrodes”. Uniformity is required to bring clarity to the manuscript.

Author response: Using foil queue microelectrode (FQ-microelectrode) for micro electrical discharge machining (micro-EDM), processed results of each foil microelectrode (F-microelectrode) can be stacked to construct three-dimensional (3D) microstructure. Therefore, FQ-microelectrode refers to foil queue microelectrode and F-microelectrode refers to foil microelectrode. And, several F-microelectrodes composed the FQ-microelectrode. Thanks for your useful comment and apologize for this again.

Round 2

Reviewer 2 Report

The authors show great effort in the improvement of the revised manuscript and in its organization. However, I suggest once again to check uniformity of language about "F-Electrodes" and "FQ-electrodes". Moreover, I suggest to read again the text, since some typos are still present, as well. Concerning the substitution of the old Figure 5, I understand the emergency in current times. I let to the editor the final decision about the chance of replacing low quality figures in the future.